# The Link between Gut Microbiota and Hepatic Encephalopathy

**DOI:** 10.3390/ijms23168999

**Published:** 2022-08-12

**Authors:** Sung-Min Won, Ki Kwang Oh, Haripriya Gupta, Raja Ganesan, Satya Priya Sharma, Jin-Ju Jeong, Sang Jun Yoon, Min Kyo Jeong, Byeong Hyun Min, Ji Ye Hyun, Hee Jin Park, Jung A. Eom, Su Been Lee, Min Gi Cha, Goo Hyun Kwon, Mi Ran Choi, Dong Joon Kim, Ki Tae Suk

**Affiliations:** Institute for Liver and Digestive Diseases, Hallym University College of Medicine, Chuncheon 24253, Korea

**Keywords:** hepatic encephalopathy, gut microbiome, gut–liver-brain axis, microbiome-targeted therapy

## Abstract

Hepatic encephalopathy (HE) is a serious complication of cirrhosis that causes neuropsychiatric problems, such as cognitive dysfunction and movement disorders. The link between the microbiota and the host plays a key role in the pathogenesis of HE. The link between the gut microbiome and disease can be positively utilized not only in the diagnosis area of HE but also in the treatment area. Probiotics and prebiotics aim to resolve gut dysbiosis and increase beneficial microbial taxa, while fecal microbiota transplantation aims to address gut dysbiosis through transplantation (FMT) of the gut microbiome from healthy donors. Antibiotics, such as rifaximin, aim to improve cognitive function and hyperammonemia by targeting harmful taxa. Current treatment regimens for HE have achieved some success in treatment by targeting the gut microbiota, however, are still accompanied by limitations and problems. A focused approach should be placed on the establishment of personalized trial designs and therapies for the improvement of future care. This narrative review identifies factors negatively influencing the gut–hepatic–brain axis leading to HE in cirrhosis and explores their relationship with the gut microbiome. We also focused on the evaluation of reported clinical studies on the management and improvement of HE patients with a particular focus on microbiome-targeted therapy.

## 1. Introduction

Hepatic encephalopathy (HE) is a major complication occurring in the cirrhosis stage, accompanied by cognitive impairment. HE is a neurological or psychiatric abnormality with a broad spectrum from minimal to overt stage as a brain dysfunction, including cognitive alterations due to hepatic insufficiency or portal systemic shunt [1]. Mild cognitive impairment resulting from HE is estimated to affect 60–80% of the population of people with cirrhosis and causes significant deterioration in daily functioning and quality of life [2,3]. The prevention and early management of HE is crucial and can improve the prognosis of affected patients [4].

The factors related to the weakening of the gut environment, such as systemic inflammation and endotoxemia, act as major mechanisms in the pathogenesis of HE [5]. Changes in the gut microbiota are a key factor in the gut environment and are closely related to the progression of cirrhosis and its complications [6]. It is unclear whether disease progresses due to the gut microbiota alone in a vast complex system, such as the gut environment; however, the study of gut microbiota will be of great help in understanding cirrhosis and subsequent HE [7]. In particular, as the impaired gut–liver-brain axis plays a fundamental role in the pathogenesis of HE, exploration of the relationship will provide new options for HE therapy [8,9].

Current management of HE consists of intestinal-centered therapy with lactulose and lactitol and nonabsorbable antibiotics, such as rifaximin and neomycin [10]. Additionally, studies are underway to apply the modulating effect of intestinal microbes through probiotics and prebiotics to the therapeutic management of HE [11]. Through this, it is being considered as another alternative as it has confirmed the possibility of improving cognitive function while suppressing harmful gut bacteria and lowering ammonia absorption [12,13]. 

In addition, although the management of HE through pharmacological approaches and nutritional methods is being studied [14,15], the link with the intestinal microflora has not been clearly identified. We explore the association of HE with the gut microbiota based on the gut–liver-brain axis and evaluate the therapeutic management of HE with a focus on clinical research data. Through this, we review the future directions by identifying the link with the gut microbiota.

## 2. Gut Microbiota

The human digestive tract contains a microbial population that is approximately ten times the number of cells in the body and forms a complex ecosystem by forming a symbiotic relationship with the host [16]. The gut microbiota matures the host’s immune system and maintains homeostasis [17]. It also performs important functions related to metabolism, including vitamin synthesis, nutrient digestion, drug action and metabolism [18].

A major issue of study related to the gut microbiota has focused on its relationship to host health and disease. Existing studies have confirmed that the gut microbiota affects the improvement or exacerbation of liver disease. The evidence related to the deterioration of gut function in patients with cirrhosis and changes in the gut microbiota caused by dysbiosis was studied. Factors, such as decreased intestinal motility, small intestinal bacterial overgrowth and decreased bile acids caused by impaired enterohepatic circulation mainly affect the gut microbiota and alter the gut environment of patients [19,20]. Animal models and clinical studies have confirmed alterations in the gut microbiota in patients with cirrhosis and HE, resulting in increased production of ammonia and endotoxins [21,22]. 

A significant increase in *Escherichia coli* and *Staphylococcus* spp. was confirmed in the gut microbiome of cirrhotic patients with minimal HE (MHE), and evidence of a disruption of the gut microbiota was also reported [12]. A similar pattern has been observed for dysbiosis in these cirrhosis conditions, and some researchers have devised and used several indices for the diagnosis and prognosis of cirrhosis. One index that measures the degree of dysbiosis in cirrhosis is the Cirrhosis Dysbiosis Ratio (CDR) [23]. This index reported by Bajaj is calculated as the ratio of the abundance of *Lachnospiraceae*, *Ruminococcaceae*, *Veillonellaceae* and *Clostridiales Incertae sedis* XIV to that of *Bacteroidaceae* and *Enterobacteriaceae*. 

This index also correlates with endotoxins and represents negative ecological and gut functional effects. Another index is the Gut Dysbiosis Index (GDI) reported by Wang [24]. The higher this index, the more severe dysbiosis appears. In addition, recently Bajaj et al. [25] reported that the addition of a microbiota index to an RNA or DNA model in the stool analysis of patients could be significantly added to the MELD score to predict the risk of hospitalization for liver cirrhosis. 

Current HE therapies are based on modulation of this gut microbiota and are aimed at reducing blood ammonia and endotoxin and improving cognitive function. To improve and improve the effectiveness of treatment for HE, it is essential to understand the changes in the composition and functions of the gut microbiota. To this end, the resolution and management of dysbiosis can be key to the therapeutic approach of the disease. Utilization of the indices mentioned can help with this approach.

## 3. Hepatic Encephalopathy

HE is a reversible syndrome with impaired brain function observed in patients with advanced liver failure or decompensated cirrhosis stages. In addition, cases of HE have been reported in patients with extensive portosystemic shunting despite the absence of cirrhosis [26,27]. HE affects the brain due to the accumulation of neurotoxic substances in the bloodstream and presents a variety of neuropsychiatric abnormalities [28]. The patient develops personality changes, disorientation and depression, and severe disease progression can eventually lead to a coma and ultimately death [29]. Although the mechanism of brain dysfunction in liver failure has not been elucidated, it is clear that it is directly related to liver failure through reduced ammonia metabolism and loss of liver function [30,31].

The main triggers for HE are infection, alcohol consumption, drug noncompliance and kidney failure, which can occur in any situation present in chronic liver disease [32]. However, the pathogenesis of HE is still not well understood. There are clear limitations in directly studying the brains of HE patients, and the existing models representing HE are not perfect. Nevertheless, it is suggested that neurotoxic effects, neurological disorders due to metabolic changes induced by liver failure, systemic inflammatory responses, alterations in the brain barrier and changes in the brain energy metabolism may play a role [33,34]. The development of HE can be triggered or progressed through the interaction and simultaneous action of several complex factors.

Neurotoxic substances presumed to affect HE include short-chain fatty acids, manganese, γ-aminobutyric acid (GABA) and false neurotransmitters, such as tyramine and octopamine, among which ammonia is the most well-known [35]. Ammonia is continuously produced in human organs as a by-product of the catabolism of amino acids. In a healthy person, ammonia is produced by gut microbiota and is converted to glutamine by the liver, thus preventing the penetration of ammonia into the systemic circulation [36]. In the liver, ammonia is converted to urea through the urea cycle in periportal hepatocytes, which is then excreted or eliminated through the urine and colon [37]. 

The urea cycle undergoes five major steps and requires enzymes, including N-acetyl-glutamate synthase, carbamoyl phosphate synthetase, ornithine transcarbamylase, argininosuccinate synthetase, argininosuccinic acid lyase and arginase [38]. Among them, glutamine synthase is present in periportal hepatocytes and prevents the release of residual ammonia into the systemic circulation [39]. This enzyme synthesizes glutamic acid and ammonia and converts them to glutamine. However, in cirrhosis or equivalent liver failure, ammonia clearance decreases and progresses to hyperammonemia [40]. Additionally, when portal hypertension develops, various portal-systemic shunts occur as a complication, which is frequently observed in cirrhosis patients [41,42]. 

These portal-systemic shunts play a role in lowering portal pressure; however, through this, normal liver flow is bypasses, resulting in increased blood concentrations without detoxification or metabolization of various toxic substances, including ammonia [43,44]. When ammonia, which is excessively accumulated in the blood, passes through the blood–brain barrier, a neurotoxic effect is expressed, which results in the swelling of astrocytes [45,46]. As astrocytes metabolize the infiltrated ammonia and produce glutamine, the osmolarity increases, which can lead to brain edema [47]. Ammonia additionally inhibits excitatory and inhibitory postsynaptic potential generation, negatively affecting neuroelectric activity and leading to impaired energy and amino acid metabolism [48] (Figure 1).

The gut microbiota is known as one of the factors that can trigger HE; however, the detailed mechanisms are not fully understood. However, intestinal changes due to dysbiosis contribute to MHE [49]. In the intestines of patients with cirrhosis and liver failure, changes occur in the abundance of pathogenic gut microbiota from overgrowth of small intestine bacteria [50]. An increase in gut microbiota taxa, such as *Porphyromonadaceae*, is positively correlated with cognitive impairment [23]. In this way, analysis of the gut microbiota from patients’ feces and saliva could be helpful in the diagnosis of HE [51].

## 4. Linkage of Gut Microbiota with Hepatic Encephalopathy

Recent cumulative data reveal a close correlation between HE and the gut microbiome [52,53]. Advances in gut microbiome analysis are providing new perspectives in the treatment of HE [54,55]. Newly developed analytical methods have moved beyond simply cataloging the composition of the gut microbiota to understanding their interrelationships with diseases. In this way, we influence our understanding of HE-targeted gut microbiota therapy, diagnosis and management. We focused on clinical data to illuminate the relationship between the gut microbiota and HE.

### 4.1. Gut Microbiota in Hepatic Encephalopathy Diagnosis

The diagnosis of HE ranges from clinical scales to psychometric and neurophysiological approaches. Testing and diagnosis options may vary depending on the severity of HE symptoms [56]. Representative symptoms of overt HE (OHE) include disorientation and asterixis [57]. In OHE, the diagnosis of cognitive impairment is not difficult to make through clinical observations and neurophysiological examinations, if other causes of mental status abnormalities, such as drug and alcohol abuse as well as hyponatremia are excluded [58]. MHE is frequently seen in patients with chronic liver disease, and brain dysfunction can be judged as test-dependent or clinical signs [58]. 

MHE does not show cognitive signs or other clinical signs and should be tested with psychometric and neurophysiological approaches [59]. MHE may cause socioeconomic difficulties due to a decrease in the patient’s quality of life [60,61]. Examination of patients with all potential risks is required for the diagnosis of MHE; however, it may cause economic difficulties due to the cost of the examination [62]. Since the tests and clinical measures currently applied to diagnose HE are not valid for the overall HE spectrum [46], adding additional diagnostic strategies should increase the reliability and determine the severity of cognitive impairment.

The close correlation between disease and the gut microbiome was the main focus of chronic liver disease study [63]. In the early days of the study of the gut microbiome of liver cirrhosis patients, the definition of changes in the gut microbiome by culture-based study was the main focus; however, after the metagenomic revolution [64], the progress of culture-independent technology made it possible to analyze the overall changes in the com-position and function of the microbiome in detail [7]. These changes in analysis technology are gradually enabling the expansion of the diagnostic field of liver diseases through the gut microbiome. Table 1 provides a summary of studies evaluating the gut microbiome in cirrhosis and HE patients as examples that can be used for diagnostic application.

Certain changes in the gut microbiome have been clearly identified between cirrhosis patients without cognitive impairment and cirrhosis patients with MHE or OHE [65]. Sung and colleagues profiled dynamic changes in the gut microbiome of cirrhotic patients with OHE compared to healthy individuals and cirrhosis patients. In patients with acute HE, the relative abundance of *Bacteroidetes* decreased while the relative abundance of *Firmicutes*, *Proteobacteria*, *Actinobacteria* and *Veillonella parvula* increased. In addition, the association between HE recurrence and survival time of specific taxa was confirmed [66]. 

Zhang et al. reported that *Streptococcus salivarius* showed a higher relative abundance in cirrhotic patients with MHE than in non-MHE cirrhotic patients. *Streptococcus salivarius* showed a positive correlation with ammonia accumulation and could be a potential biomarker targeting MHE cirrhotic patients [67]. Bajaj et al. confirmed the difference in the composition of fecal microbiota between healthy controls and cirrhosis patients and analyzed the change in the fecal microbiota of cirrhosis patients according to the presence or absence of HE [68]. 

The fecal microbiota of cirrhotic patients were significantly different from those of healthy controls. The relative abundances of *Enterobacteriaceae*, *Alcaligeneceae* and *Fusobacteriaceae* were high but lower for *Ruminococcaceae* and *Lachnospiraceae*. High levels of *Veillonellaceae* and endotoxemia were found between cirrhosis patients without HE and HE patients, and in particular *Alcaligeneceae* and *Porphyromonadaceae* were reported to have a positive correlation with cognitive impairment. In addition, since *Alcaligenaceae* can produce ammonia through urea decomposition, the association of HE with cognitive impairment was confirmed. 

Subsequently, the same researchers successively analyzed the microbiome of the feces and colonic mucosa of cirrhosis patients [49]. The control group had lower pathogenic bacterial genera than cirrhotic patients, including HE in the microbiome, and HE patients had a poorer model for end-stage liver disease (MELD) score and higher endotoxin levels than no-HE patients. Although the microbiome analyzed in the stool was not prominent, the sigmoid mucosal microbiome showed a significant difference, further confirming the difference with or without HE. 

In the mucosal microbiota of HE patients, *Enterococcus*, *Veillonella*, *Megasphaera* and *Burkholderia* predominated at high levels, whereas *Roseburia* showed low levels compared to the control group. *Blautia*, *Fecalibacterium*, *Roseburia* and *Dorea* were associated with good cognition and reduced inflammation in both HE and no-HE patients, whereas *Enterococcus*, *Megasphaera* and *Burkholderia*, prominent in HE patients, were strongly associated with cognitive impairment.

A clear difference in the gut microbiome between the healthy control group and cirrhosis patients with liver cirrhosis and HE was revealed through various lines of evidence, and it was confirmed that these differences were strongly related to symptoms, such as cognitive impairment and inflammation. This difference is expected to bring accuracy to the diagnosis of the disease and will act as a clue in the therapeutic stage.

**Table 1 ijms-23-08999-t001:** Human study of microbiome change in cirrhosis with hepatic encephalopathy.

Author	Scheme	Group	Result	Ref.
Sung et al., 2019	Profiled fecal microbiome changes for a cohort patients	Control/compensated cirrhosis/decompensated cirrhosis/Acute HE	Higher *Firmicute*, *Proteobacteria*, *Actinobacteria* and *Veillonella parvula* and lower *Bacteroidetes* phylum in AHE patients compared compensated cirrhosis.	[66]
Zhang et al., 2013	Gut microbiome analysis of MHE patients with cirrhosis	MHE/cirrhotic patients without MHE	Higher *Streptococcus salivarius* in MHE cirrhotic patients compared no-MHE cirrhotic patients. Gut ammonia-increasing bacteria *Streptococcus salivarius* can be a potential biomarker in MHE cirrhotic patients.	[67]
Bajaj et al., 2012	Stool analysis of cirrhosis patients and age-matched controls	Cirrhotic patients with or without HE/control	Higher *Veillonellaceae* in HE cirrhotic patients. *Alcaligenaceae* and *Porphyromonadaceae* were positively correlated with cognitive impairment in cirrhotic patients	[68]
Bajaj et al., 2012	Sigmoid mucosal and fecal microbiome analysis to study linkage with cognition and inflammation	OHE patients/no-OHE patients/control	Higher *Enterococcus*, *Veillonella*, *Megasphaera* and *Burkholderia* in mucosal microbiota of HE patients. Lower *Roseburia* in mucosal microbiota of HE patients. *Blautia*, *Fecalibacterium*, *Roseburia* and *Dorea* are associated with a positive cognition state and *Enterococcus*, *Megasphaera* and *Burkholderia* are associated with a poor cognition state.	[49]

AHE, acute episode of overt hepatic encephalopathy; HE, hepatic encephalopathy; MHE, minimal hepatic encephalopathy; and OHE, overt hepatic encephalopathy.

### 4.2. Gut Microbiota in HE Treatment

Given the impact and key role of the microbiota and host interaction on HE, it is only natural that the direction of the investigation of HE therapy targets the gut microbiota. Several treatments have been studied, ranging from direct approaches aimed at relieving dysbiosis observed in cirrhosis patients or reducing harmful taxa, to targeting metabolites by microbiota. Here, we refer to several therapeutic approaches and summarize the cases.

#### 4.2.1. Probiotics and Prebiotics

The various effects of probiotics are being proven or tested in a number of diseases. Likewise, probiotics are being considered as one of the treatments that can improve HE and have been applied and studied in several studies. The expected effects of probiotics include improvement of intestinal barrier function by modulating gut microbiota, immune regulation and HE treatment effects by reducing portal hypertension. In addition, synbiotics are being studied as potential therapeutics for HE and lactulose, a representative nonabsorbable disaccharide and prebiotic, is being used as an effective treatment for HE (Table 2).

A previous study evaluated the effectiveness of *Lactobacillus rhamnosus* GG (LGG) in patients with MHE in a phase 1 randomized controlled trial [52]. Patients were treated with LGG or placebo for 8 weeks, and the fecal microbiota, inflammation and endotoxin levels were analyzed. As a result, *Enterobacteriaceae* decreased in the LGG group, the relative abundance of *Clostridiales* Incertae Sedis XIV and *Lachnospiraceae* increased, and endotoxemia and TNF-α decreased. However, as the efficacy of the study was not proven, no cognitive changes were observed between patients. 

Xia et al. conducted a probiotic-based treatment for HBV cirrhosis patients with MHE [69]. Probiotics were given to patients in combination with *Clostridium butyricum* and *Bifidobacterium infantis*. In the intestinal microflora of the patients, *Clostridium* cluster I and *Bifidobacterium* were increased and *Enterococcus* and *Enterobacteriaceae* were decreased, which also decreased the level of venous ammonia and had an effect on cognitive improvement. In another study, a mixture of VSL#3 probiotics was administered to cirrhotic patients without HE and their effectiveness was confirmed [70]. Reductions in arterial ammonia levels, small intestinal bacterial overgrowth (SIBO) and orocecal transit time (OCTT) were observed after 3 months of probiotic treatment. 

As a result, it was possible to confirm the primary prevention effect in OHE through the administration of probiotics. Another study confirmed the effect of synbiotics in MHE patients without OHE through an additional combination [12]. The synbiotics used in the study were provided to patients by combining urease-free bacteria and fermentable fibers. As a result, the content of *Lactobacillus*, which does not produce urease, was increased in the patient’s feces, and endotoxemia was reduced. 

In addition, improvement in the Child–Turcotte–Pugh functional class, reduction in blood ammonia levels and reversal of MHE were observed in 50% of patients receiving synbiotics. Other researchers who provided patients with a combination of *Bifidobacterium* and fructooligosaccharides reported improvements in psychometric tests compared to the lactulose-treated group, with improvements in blood ammonia levels and psychometric tests after 60 days.

Lactulose is a synthetic, nonabsorbable disaccharide used as one of the treatments for HE that affects the gut microbiota and reduces the absorption of ammonia. In a study evaluating the effect of lactulose on MHE patients, a reduction in bacterial DNA translocation and improvement in neurocognitive test scores were observed [71]. This is likely due to the effect of lactulose in changing the gut microbiota and improving intestinal permeability. In a randomized controlled trial in HE patients, positive alterations in the gut microbiota were observed in patients responding to lactulose treatment [72]. Significant differences were observed between lactulose responders and nonresponders in *Bacteroidetes*, *Firmicutes*, *Actinobacteria* and *Proteobacteria*.

However, the limitations as therapeutic agents for the abovementioned probiotics and prebiotics are still clear. Accurate comparative analysis is difficult due to differences in research design and delivery methods of probiotics used in each clinical study. In addition, the question remains as to whether the intestine of cirrhosis patients can be reconstituted into a healthy complex microbiome through probiotics or prebiotics as an environment in which dysbiosis occurs [73]. 

Probiotics, prebiotics and synbiotics may not reach the vulnerable intestine of cirrhotic patients correctly or may be eliminated by antibiotic treatment and may not induce changes in the intestinal environment. Lactulose, which is widely used as a treatment for HE, was also reported to have increased bacterial imbalance despite lactulose treatment in HE patients, unlike the previously reported effects of inducing changes in the gut microbiota [23]. 

In addition, the evaluation of the therapeutic effect of HE on nonabsorbable disaccharides remains controversial. A meta-analysis reported in the Cochrane Review in 2004 found that nonabsorbable disaccharides had a beneficial effect on HE but not on mortality. However, in the review at the time, the argument of the conclusion about nonabsorbable disaccharide showed weakness due to the lack of statistical power and methodological problems, such as bias area reporting. 

Subsequently, an updated 2016 Cochrane Review found evidence for the efficacy and safety of nonabsorbable disaccharides for the treatment and prevention of HE in cirrhosis patients and reached conclusions. The use of nonabsorbable disaccharides has significantly confirmed minimal and overt therapeutic effects for HE and has an overall beneficial effect on liver-related morbidity and all-cause mortality. 

The evaluation of nonabsorbable disaccharides has been revisited with the ongoing development of new randomized clinical trials and updates to the European Association for the Study of the Liver (EASL) and the American Association for the Study of Liver Diseases (AASLD) practice guidelines. In order to evaluate the controversies of these microbial-targeted therapies, improve the limitations and increase the therapeutic effects, alternatives, such as data accumulation through large-scale studies and establishment of standards should be prepared.

**Table 2 ijms-23-08999-t002:** Probiotics and prebiotics therapies for hepatic encephalopathy.

Author	Scheme	Group	Result	Ref.
Bajaj et al., 2014	A randomized clinical phase 1, placebo-controlled trial	LGG or placebo in cirrhotic patients with MHE	In the LGG group, *Enterobacteriaceae* decreased and the relative abundance of *Clostridiales Incertae Sedis XIV* and *Lachnospiraceae* increased. Endotoxemia and TNF-α were decreased in the LGG group but there was no change in cognitive function.	[52]
Xia et al., 2018	A randomized clinical trial	Probiotics (*Clostridium butyricum* combined with *Bifidobacterium infantis*) or no probiotics	*Clostridium* cluster I and *Bifidobacterium* increased and *Enterococcus* and *Enterobacteriaceae* decreased in the group treated with probiotics in HBV cirrhosis patients with MHE. There was also a reduction in venous ammonia and cognitive improvement.	[69]
Lunia et al., 2014	A prospective, randomized controlled trial	Cirrhosis patients without OHE given probiotics (VSL#3)/cirrhosis patients without OHE not given probiotics (VSL#3)	Treatment with probiotics for 3 months significantly reduced arterial ammonia, SIBO and OCTT levels. Probiotics have shown a preventing effect on HE.	[70]
Liu et al., 2004	A randomized clinical trial	Synbiotic preparation/fermentable fiber alone/placebo	Synbiotic treatment increased the fecal content of *Lactobacillus* species that do not produce urease. Modulation of the gut microbiota showed a reduction in blood ammonia levels and reversal of MHE in 50% of patients.	[12]
Malaguarnera et al., 2010	A randomized controlled trial	*Bifidobacterium* + fructooligosaccharides or lactulose in HE patients	The combination of *Bifidobacterium* and fructooligosaccharide at 30 days of treatment showed improvement in psychometric tests compared to the lactulose group. Treatment for 60 days showed significant improvement in psychometric tests and blood ammonia levels.	[74]
Moratalla et al., 2017	Observational cohort study of cirrhosis patients with MHE	First cohort: MHE patients with or without lactulose Second cohort: Non-lactulose MHE patients going to initiate lactulose therapy	Lactulose reduces bacterial DNA translocation and improves neurocognitive test scores in MHE patients.	[71]
Wang et al., 2019	A multicenter, open-label randomized controlled trial	Lactulose or control in cirrhotic patients	Treatment with lactulose significantly improved MHE recovery. Significant differences were found between lactulose responders and non-responders in *Bacteroidetes*, *Firmicutes*, *Actinobacteria* and *Proteobacteria*.	[72]

HE, hepatic encephalopathy; LGG, *Lactobacillus rhamnosus* GG; MHE, minimal hepatic encephalopathy; OCTT, orocecal transit time; OHE, overt hepatic encephalopathy; and SIBO, small intestinal bacterial overgrowth.

#### 4.2.2. Fecal Microbiota Transplantation

Fecal microbiota transplantation (FMT) is the transfer of stool from a ‘healthy’ donor to a patient with an imbalanced gut microbiota for recovery purposes [75]. When FMT is performed in HE patients, it can be expected to change the composition of the gut microbiota to rebuild intestinal barrier integrity and reduce ammonia absorption (Table 3). In several animal models, FMT had confirmed effects in reducing intestinal ammonia production in the gut and decreasing the reduced risk of encephalopathy [76].

In a case report, Kao et al. demonstrated for the first time that continuous FMT in patients with mild HE improved cognitive function as assessed by the Stroop test and inhibitory control test [77]. However, the beneficial effects did not persist after FMT was discontinued, and there were reports of results suggesting that it is unclear whether FMT is effective for severe HE. Bajaj et al. performed FMT in patients with recurrent HE and compared it with the standard of care [54]. For rational donor selection, feces from donors with the highest relative abundance of *Lachnospiraceae* and *Ruminococcaceae* were selected using machine-learning techniques. 

At the 5-month follow-up, compared with standard of care, improvement of dysbiosis and improvement of cognitive function were confirmed through an increase in beneficial taxa and diversity. Afterward, similar results were reported in phase 1 clinical trials using FMT oral capsules in patients with recurrent HE [78]. Oral FMT capsules were prepared from the feces of the same donor as above. The patients who received the FMT capsule showed improvement in the cognitive cognition test using EncephalApp but not in the test using the psychometric HE score. However, the microbiome diversity and intestinal microbiome imbalance of the duodenal mucosa were improved, and the expression of antimicrobial peptides and reduction in lipopolysaccharide-binding proteins were confirmed.

One of the key factors to consider first in FMT is the optimization of donors and recipients. Bloom et al. [79] evaluated the safety and efficacy of FMT in patients with overt HE through donor diversity. A total of five FMT donors were screened and administered fivefive FMT capsules over 3 weeks. After 6 months of follow-up of 10 patients, no change in the Model for End-Stage Liver Disease (MELD) score was observed. In the evaluation of serious adverse events in this study, 13 minor adverse events and one extended-spectrum beta-lactamase Escherichia coli bacteremia were reported. 

After FMT, the psychometric HE score improved, and the range of change varied among donors. In addition, high levels of *Bifidobacterium* and other known beneficial taxa were identified in stool analysis of FMT responders. In conclusion, cognitive improvement in HE was confirmed; however, the effect varied widely between donors and recipients. The study reported by Bloom et al. [79] is an important reminder of the importance of donor and recipient factors and the importance of selection.

Overall, studies evaluating the effects of FMT on HE have been designed with safety-first objectives, and definitive efficacy evaluations are lacking. In addition, there are risk factors for infection and safety due to the lack of standards and tests for donor selection. The potential benefits of FMT are clear; however, large-scale studies are needed to establish additional dosing strategies.

#### 4.2.3. Antibiotics

Antibiotics in the treatment of HE are suggested as a way to deplete certain taxa in the gut that produce urease-producing bacteria or neurotoxins and reduce the systemic immune response [80]. It is known that certain antibiotics can potentially survive beneficial taxa and selectively inhibit harmful taxa [81]. The current mainstream treatment for HE is a combination of antibiotics and lactulose (Table 4). Rifaximin, recently used to treat HE, is an antibiotic with broad-spectrum activity against both Gram-positive and Gram-negative, aerobic and anaerobic bacteria [82]. 

Rifaximin is an antibiotic approved in the United States and Europe to reduce the risk of recurrent OHE [1]. Administration of rifaximin to HE patients reduced hyperammonemia and endotoxemia and showed cognitive improvement [83]. In current clinical practice, rifaximin is recommended as an additional therapy to prevent OHE recurrence and is prescribed in combination with lactulose for OHE patients [83]. Several clinically controlled trials have reported that rifaximin significantly reduces the risk of recurrent OHE and improves the cognition and quality of life in MHE patients [84]. 

The effectiveness of rifaximin was also confirmed in a study that determined whether rifaximin was associated with the risk of death and cirrhosis complications. Kang et al. [85] reported that the risk of death and the reduced risk of recurrent HE were significantly related in the group of patients receiving rifaximin and lactulose compared to the control group receiving only lactulose in non-hepatocellular carcinoma (HCC) cohort. In this way, rifaximin treatment significantly affects the prolongation of overall survival and reduced risk of complications, such as spontaneous bacterial peritonitis, varicose bleeding and recurrent hepatic encephalopathy [85].

A decrease in endotoxin activity and serum ammonia levels was reported in 20 patients with decompensated cirrhosis after 4 weeks of rifaximin administration [86]. In addition, it was confirmed that cognitive improvement was significantly achieved through this. A decrease in the genera *Veillonella* and *Streptococcus* was observed; however, the effect of rifaximin on the diversity of the gut microbiome was not confirmed. Another clinical study in which rifaximin was administered to 20 MHE patients also confirmed a significant improvement in endotoxemia [87]. 

After rifaximin was administered, cognitive improvement was confirmed, and an increase in serum saturated and unsaturated fatty acids was observed. There was a decrease in *Veillonellaceae* and an increase in *Eubacteriaceae*; however, there was no significant change in the overall intestinal microflora. However, a change was confirmed in the correlation network analysis with rifaximin. In networks centered on *Enterobacteriaceae*, *Porphyromonadaceae* and *Bacteroidaceae*, transfer changes associated with association with beneficial metabolites and cognitive improvement were observed.

Other researchers have confirmed the gut microbiota and therapeutic effects of rifaximin alone or rifaximin plus probiotics administered to MHE patients [88]. In both treatment groups, a decrease in the diversity of the gut microbiota was observed, and a decrease in certain ammonia-producing taxa, such as *Clostridium* was observed. Rifaximin showed a decrease in *Streptococcus*, *Veillonella* and *Lactobacillus* with hyperammonemia and cognitive improvement, however, still showed no significant effect or change on the overall relative abundance of bacteria [89,90]. In patients with cirrhosis and MHE, rifaximin alone or rifaximin plus lactulose consecutively for 3 months did not show any significant changes in the microbiome [91]. The microbiota analyzed in duodenal samples and feces showed clear differences; however, no changes were observed before versus after treatment.

Combining the reported clinical results, the effect of rifaximin on the overall gut mi-crobiota composition has not been observed and no conclusion has been reached either. Detailed changes in species or subgroups have not been measured and may not have confirmed the effect of such rifaximin. The effect of rifaximin on changes in the composition of the gut microbiome has been demonstrated in animal studies and in changes in microbial metabolites, such as bile acids [92,93]. Changes in beneficial taxa in irritable bowel syndrome or Crohn’s disease are clearly observed [94]. To analyze the effects and mechanisms of rifaximin on the intestinal microflora for the treatment of HE in the future, it is necessary to identify changes at the species and subtaxa levels.

**Table 4 ijms-23-08999-t004:** Antibiotics therapies for hepatic encephalopathy.

Author	Scheme	Group	Result	Ref.
Kang et al., 2017	A retrospective cohort study	Rifaximin + lactulose or lactulose in non-HCC cohort or HCC cohort	In the non-HCC cohort, rifaximin was significantly associated with a lower risk of death and reduced the risk of recurrent HE, spontaneous bacterial peritonitis. In the HCC cohort, rifaximin was not associated with a risk of death. It was associated with a lower risk of spontaneous bacterial peritonitis but not with varicose bleeding or recurrent HE. The risk of *Clostridium difficile*-associated diarrhea was not different between the two groups.	[85]
Kaji et al., 2017	A clinical trial	20 patients with decompensated cirrhosis (Child–Pugh score > 7)	Treatment with rifaximin for 4 weeks resulted in a decrease in endotoxin activity and serum ammonia levels. Treatment with rifaximin for 4 weeks resulted in a decrease in endotoxin activity and serum ammonia levels. There was no significant difference in the diversity and composition of gut microbiota at baseline and after 4 weeks of treatment but the relative abundance of genus *Veillonella* and *Streptococcus* was lowered.	[86]
Bajaj et al., 2013	A clinical trial	20 patients with cirrhosis who had been diagnosed with MHE	There was a significant cognitive improvement and a decrease in endotoxemia after rifaximin treatment. Serum saturated and unsaturated fatty acids were significantly increased after rifaximin treatment. No significant changes in gut microbiota were observed except for the decrease of *Veillonellaceae* and the increase of *Eubacteriaceae*.	[87]
Zuo et al., 2017	A randomized clinical trial	Rifaximin or rifaximin and probiotics in cirrhotic patients with MHE	Both treatments reduced overall microbiome diversity and decreased abundance of specific ammonia-producing bacteria. The treatment with rifaximin + probiotics showed a more definite effect. Patients with nonalcoholic MHE were more responsive to microbiota alteration therapy.	[88]
Suzuki et al., 2018 Kawaguchi et al., 2019	A prospective, randomized studies (a phase II/III study and a phase III study)	Rifaximin or lactitol with grade I or II HE and hyperammonemia patients	Blood ammonia concentrations were significantly improved in the rifaximin group. The portal systemic encephalopathy index was significantly improved in both groups. As a result of fecal microbiome analysis of 17 participants in the clinical trial, the number of *Streptococcus*, *Veillonella* and *Lactobacillus* decreased after rifaximin treatment. Rifaximin alters the composition of microbial taxa linked to hepatic/neuropsychological function.	[89,90]
Schulz et al., 2019	A randomized clinical trial	Rifaximin or rifaximin and lactulose in cirrhotic patients with MHE	An improvement in MHE was confirmed after treatment. Microbiological analysis was performed on duodenum and stool samples and no statistically significant changes were found in the bacterial profile. Rifaximin therapy with or without lactulose for 3 months has no effect on microbiome composition. HE improvement with rifaximin persisted after termination.	[91]

HCC, hepatocellular carcinoma; and MHE, minimal hepatic encephalopathy.

#### 4.2.4. Dietary

Therapeutic strategies for HE have been to modulate the systemic inflammation, neurotoxic substances, such as ammonia, and changes in the composition and environment of the gut microbiota [95]. Additionally, implementing these strategies in conjunction with a diet at the same time can play an important role in treatment and management [96]. It has been known through many reports that changes in eating habits are closely related to the gut microbiota [97]. These dietary adjustments and changes have beneficial effects on the gut microbiome and affect disease improvement. Likewise, it can alter the cascades that lead to cognitive impairment by regulating the gut nitrogen metabolism and improving inflammation [98]. 

For the management of HE, recommendations for dietary changes are presented as key clinical guidelines, which can make a significant contribution to treatment and prevention. The American Association for the Study of Liver Diseases (AASLD), in its 2014 guidelines [1], recommended a calorie intake of 35–40 kcal/kg/day for patients with cirrhosis and HE. 

The International Society for Hepatic Encephalopathy (ISHEN) also recommends a calorie intake of 35–40 kcal/kg/day [11], and the European Association for the Study of the Liver (EASL), in its latest guidelines published in 2019 [99], recommends at least 35 kcal/kg/day. Additionally, the European Society for Clinical Nutrition and Metabolism (ESPEN) recommends a calorie intake of 30–35 kcal/kg/day [99]. The four organizations recommend a protein intake of 1.2–1.5 g/kg/day. In particular, lipids in the diet are beneficial to patients with HE, as they have been shown to have beneficial effects on gut microbiota and bowel transit time [15].

As abnormal nitrogen metabolism plays an important role in the pathogenesis of HE, and early studies suggested that protein intake should be reduced [100]. However, further studies have reported that it has no beneficial effect on the progression of HE and worsens the nutritional status of patients by exacerbating protein breakdown in muscle [101]. Rather, it has been found that normal protein intake is well tolerated in HE and is useful in ensuring sufficient substrates for energy synthesis and hepatocyte function [102]. 

As additional nutrients, branched-chain amino acids, such as valine, leucine and isoleucine, have been shown to prevent excessive protein catabolism and reduce ammonia levels in patients with HE [103]. There are still obstacles and limitations in the application of these dietary and nutritional treatment modalities for HE, and high-quality randomized controlled trials are required [104]. In addition, there is still little research on the effect of diet and nutrition on the relationship between HE and the gut microbiota we are dealing with. If the positive factors of diet on this relationship are identified or proven in the future, this could be another option for HE treatment strategies.

## 5. Challenges and Limitations

The human gut microbiome is constantly exposed to external factors, such as diet, drugs and pathogens but has the ability to restore equilibrium after perturbation [105]. However, in diseases, such as cirrhosis, dysbiosis occurs in which the overgrowth of harmful taxa is promoted and beneficial taxa is suppressed [106]. Therefore, the effect of HE-related gut microbiota therapy should be confirmed through continuous iterations rather than single short-term therapy. Treatment with the gut microbiome occurs through an ecosystem connected by the gut–liver axis [107]. Due to the interconnected components of these complex ecosystems, a change in the composition of a single microbial community can have unexpected ramifications or no impact or effect at all [108].

FMT, which is being studied as one of the treatments for HE, has its own challenges and limitations [109]. As it is donated from the feces of healthy individuals, it will change over time. There are also difficulties in identifying pathogens from FMT materials. Due to this problem, there has recently been a case of infection by ESBL-producing *Escherichia coli* [110,111]. Clinical trials of FMT for HE used different routes, doses and timing of administration. Due to these differences, there is still a lack of understanding of the best FMT dosing regimen [109,112]. Criteria for an ideal FMT donor are also lacking. This optimal donor selection may require detailed analysis of microbiome composition as well as function. Additionally, there are restrictions on the preparation of FMT. When prepared aerobically, the benefits of anaerobicity are lost because of the lack of supplementation for anaerobic microorganisms.

The use of antibiotics, the main treatment for HE, also poses an important problem, as it can lead to multidrug resistance. Although the prevalence of resistant bacteria continues to rise [113,114], the use of antibiotics for the treatment of HE has no choice but to be used in a balanced way with the problem of resistance in mind.

Our review of HE addressed only a narrow category of links with the gut microbiota. Currently, pharmacological approaches to the treatment of HE are also being actively studied. L-ornithine-L-aspartate increases ammonia metabolism in the liver and muscle, thereby lowering blood ammonia levels [115,116]. Although it has been effective in improving the mental state and psychometric performance of patients in clinical studies [117], there are still problems and hurdles that are not approved in many countries. 

Sodium benzoate may be used in patients with congenital urea circulation defects and is an FDA-approved drug for the treatment of acute hyperammonemia and related encephalopathy [118,119]. However, there are problems, including the management of ascites in cirrhosis patients. This pharmacological approach could be proposed as an effective way for the treatment of HE. However, there are no studies on the relationship between these pharmacological treatments and the gut microbiome. Therefore, it is not within the scope of this review, as it departs from the solution we address in this review through modulation of the gut microbiome.

## 6. Future Directions

The treatment of HE through the gut microbiome is being achieved through major therapies, such as lactulose and rifaximin, as well as some applied therapies, such as probiotics, prebiotics and FMT. However, many HE patients still suffer from persistent symptoms. The future directions of therapeutic studies targeting the gut microbiome should be aimed at personalized treatment approaches, therapy selection and clinical trial design. 

Since therapy using the gut microbiome affects the gastrointestinal tract as a whole, it is necessary to check the effects on each part of the intestine and the effects on intestinal permeability. Further studies, including larger trials, are needed to determine the ideal dosing regimen for FMT, donor selection and the benefits of anaerobic conditions. Research into the unique gut microbiome of HE patients and therapies designed for their gut type should also be pursued. A personalized approach will be an important issue in future directions for treatment success as it can maximize the effectiveness of therapies.

For the treatment and management of HE, the scope is expanding with convergence [120] with new areas: First, as dysbiosis is induced in cirrhosis and it has been confirmed that this gut microbiota imbalance affects bacterial products and metabolites [68,121], related multiomics studies are being conducted. Researchers are focusing on the gut microbiota of metabolic potential comparable to the liver and are studying the treatment of liver disease and related complications based on the gut–liver axis [122,123]. 

In introducing the concept of ‘functional metagenomics’, Li et al. [124] demonstrated that functional key members of the gut microbiome have a significant impact on the metabolism and health of the host. Other researchers have discovered altered immune pathways through a multiplex approach to profiling of transcripts, metabolites and cytokines in blood samples from cirrhosis patients [125]. Going forward, we will go beyond generally focusing on single or target components, such as short-chain fatty acids [126] to deepen our understanding of gut microbiota dynamics in relation to host physiology and pathology [127], exploring the broad and complex host-microorganism metabolic interactions in the superorganism. 

This may provide new inspiration for the treatment of cirrhosis and HE. The second approach to new territory is AI. Recently, methods utilizing artificial intelligence and machine-learning algorithms are being applied to hepatology [120]. Machine learning is the use of artificial intelligence to create and utilize predictive models on large data sets more efficiently and effectively than traditional methods [120]. MHE can be detected by special, time-consuming psychometric tests. Recently, as the diagnostic approach of MHE through artificial intelligence and machine learning is progressing, uncertainty resolution is being studied [128]. 

Artificial intelligence and machine learning are being applied in a variety of ways, from analysis using magnetic resonance imaging data [129,130] to methods of distinguishing patients with different cirrhosis severities by the microbiome of saliva and feces [65]. In particular, the use of artificial intelligence (AI) could become a necessity in the near future, especially for the utilization of large data sets, such as linking gut microbiome data and human biometric data. 

Artificial intelligence and machine learning will help classify disproportionate medical data and build risk prediction and diagnostic evaluation systems for liver cirrhosis with HE [131]. The third is a personalized therapeutic approach. The gut microbiome has the potential to uniquely identify an individual like a fingerprint [132]. Therefore, even the same treatment may have different responses [133]. A method for determining baseline microbiome characteristics has been proposed to be consistent with an appropriate microbiome therapy. They also reported that a personalized approach to microbiome treatment based on baseline community structure and function may achieve the most clinical success [133].

The future direction of the treatment and management of cirrhosis and HE will be the link between the gut microbiota and the host based on the gut–liver axis.

## Figures and Tables

**Figure 1 ijms-23-08999-f001:**
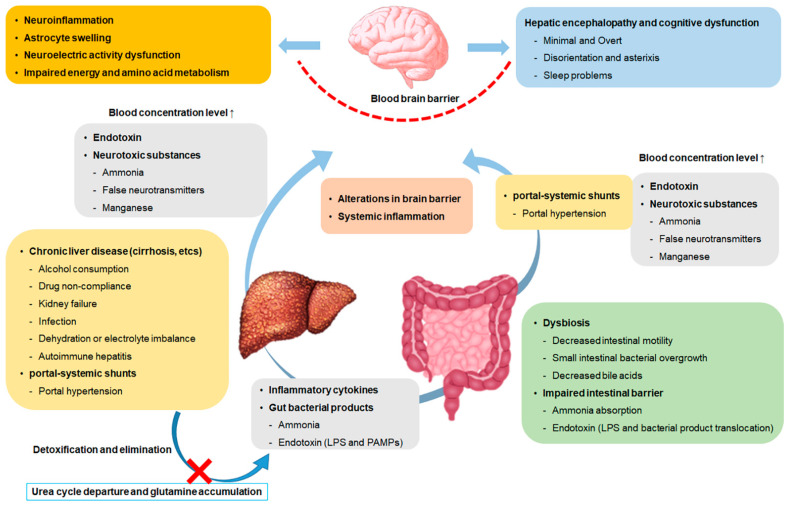
Schema of brain dysfunction by the gut-liver-brain axis. LPS, Lipopolysaccharide; and PAMPs, pathogen-associated molecular patterns.

**Table 3 ijms-23-08999-t003:** Fecal microbiota transplantation therapies for hepatic encephalopathy.

Author	Scheme	Group	Result	Ref.
Bajaj et al., 2017	An open-label, randomized clinical trial	FMT or SOC alone in Cirrhotic patients with recurrent HE on SOC	FMT resulted in cognitive improvement and reduced dysbiosis in cirrhotic patients with recurrent HE.	[54]
Kao et al., 2016	A case report	57-year-old man suffering from grade 1–2 HE with liver cirrhosis	Cognitive function improvement was confirmed as a result of evaluation of Stroop test and inhibitory control test after continuous FMT in patients with mild HE.	[77]
Bajaj et al., 2019	A randomized clinical phase 1, placebo-controlled trial	FMT capsules or placebo in Cirrhotic patients with recurrent HE	Oral FMT capsules have demonstrated safety and tolerability in patients with cirrhosis and recurrent HE. Oral FMT capsules improved duodenal mucosal diversity, dysbiosis and AMP expression and reduced serum LBP.	[78]
Bloom et al., 2022	A randomized clinical phase 2, trial	10 Overt HE patients (five FMT donors)	There was no change in MELD scores. After FMT, the psychometric HE score improved. In stool analysis of FMT responders, the levels of *Bifidobacterium* and beneficial taxa were high. 13 minor adverse events and one serious adverse event were reported. The effect varied according to the difference between donor and recipient.	[79]

AMP, antimicrobial peptide; FMT, fecal microbiota transplantation; HE, hepatic encephalopathy; LBP, lipopolysaccharide-binding protein; MELD, Model for End-Stage Liver Disease; and SOC, standard of care.

## Data Availability

Data are contained within the article.

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
