# Peer review of "The Link between Gut Microbiota and Hepatic Encephalopathy"

_ijms, 2022, doi:10.3390/ijms23168999_

Round 1

Reviewer 1 Report

POINT-BY-POINT REVISION

The link between gut microbiota and hepatic encephalopathy ” 

Won and colleagues present a clear and exhaustive review on hepatic encephalopathy (HE) focusing on its linkage with the gut microbiota. In particular, they made an excursus on the alteration in gut microbiota observed in cirrhotic patients with overt or minimal HE, on the possible role of microbiota in the pathogenesis of HE and finally on the therapeutic options of HE modyfing gut microbiota composition. 

I have few minor comments:

- In the Tables 1,2,3 and 4 I suggest to add an initial column specifing the reference (author and year of publication of the reported study) . 

- At page 3 line 120, please change the sentence “ if mental status abnormalities such as drug, alcohol abuse and hyponatremia are excluded” in “ if other causes of mental status abnormalities such as drug, alcohol abuse and hyponatremia are excluded”

-At page 3 line 125 “MHE may cause socioeconomic difficulties due to a decrease in the patient's quality of life” please add the references: Bajaj JS, Riggio O, Allampati S, Prakash R, Gioia S, Onori E, Piazza N, Noble NA, White MB, Mullen KD. Cognitive dysfunction is associated with poor socioeconomic status in patients with cirrhosis: an international multicenter study. Clin Gastroenterol Hepatol. 2013 Nov;11(11):1511-6 and Ridola L, Nardelli S, Gioia S, Riggio O. Quality of life in patients with minimal hepatic encephalopathy. World J Gastroenterol. 2018;24:5446–53.

Author Response

“The link between gut microbiota and hepatic encephalopathy”

Point-to-point responses to comments by the Reviewer 1

First of all, we would like to thank the Reviewer 1 for his/her comments, which helped us to improve this manuscript.

Specific Comments:

Won and colleagues present a clear and exhaustive review on hepatic encephalopathy (HE) focusing on its linkage with the gut microbiota. In particular, they made an excursus on the alteration in gut microbiota observed in cirrhotic patients with overt or minimal HE, on the possible role of microbiota in the pathogenesis of HE and finally on the therapeutic options of HE modyfing gut microbiota composition.

I have few minor comments:

  • Comment 1: In the Tables 1,2,3 and 4 I suggest to add an initial column specifing the reference (author and year of publication of the reported study) .
  • Response 1: Thanks for these suggestions. We have added and edited the part you mentioned.
  • Comment 2: At page 3 line 120, please change the sentence “if mental status abnormalities such as drug, alcohol abuse and hyponatremia are excluded” in “ if other causes of mental status abnormalities such as drug, alcohol abuse and hyponatremia are excluded”
  • Response 2: As reviewer suggested, we have edited the part you mentioned
  • Comment 3: -At page 3 line 125 “MHE may cause socioeconomic difficulties due to a decrease in the patient's quality of life” please add the references: Bajaj JS, Riggio O, Allampati S, Prakash R, Gioia S, Onori E, Piazza N, Noble NA, White MB, Mullen KD. Cognitive dysfunction is associated with poor socioeconomic status in patients with cirrhosis: an international multicenter study. Clin Gastroenterol Hepatol. 2013 Nov;11(11):1511-6 and Ridola L, Nardelli S, Gioia S, Riggio O. Quality of life in patients with minimal hepatic encephalopathy. World J Gastroenterol. 2018;24:5446–53.
  • Response 3: We checked and added the reference you mentioned.

Reviewer 2 Report

Generally said, the manuscript gives a well organized overview about alterations of gut microbiota in patients with liver cirrhosis, the contribution of dysbiosis to HE and the potential therapeutic effects of modulating dysbiosis on manifestation of HE.

My major point of criticism concerns the paragraphs describing the pathogenesis of HE, and particularly the hyperammonemia. The main part of blood ammonium originates in the gut and is produced by intestinal bacteria. In dysbiosis, ammonia poduction may be increased, and endotoxin production has a lot of damaging effects.

In addition, due to the emergence of porto-systemic collaterals, part of ammonia bypasses the liver (not mentioned in the paper!). Further point: Ammonia clearence by the liver is impaired (urea cycle in periportal hepatocytes, glutamin synthesis in perivenous cells. This an important nechanism, not mentioned in the paper. Considering these facts, Figure 1 must be improved.

Next point: In Fig. 1, the most important causes of liver cirrhosis of chronic liver disease are mentioned. However, metabolic and autoimmune liver diseases are forgotten. Please correct.

The presentation of studies about the link between gut dysbiosis and HE seems to be exhaustive. Maybe, it could be shortened. In page 4, several studies or authors are mentioned, but the citation lacks.

I know that the authors focus on gut dysbiosis. Therefore, multiple studies are indicated that are potentially effective in treatment of HE, mainly by by reducing hyperammonemia. However, the authors don't mentione other measures to lower ammonium levels, e.g. branched chain aminoacids, or L-ornithin-L-aspartate or similar compounds.

My question: Can nutrition alter dysbiosis in liver cirrhosis? The authors may comment on that.

Author Response

ijms-1826515

“The link between gut microbiota and hepatic encephalopathy”

Point-to-point responses to comments by the Reviewer 2

First of all, we would like to thank the Reviewer 2 for his/her comments, which helped us to improve this manuscript.

Specific Comments:

  • Comment 1: My major point of criticism concerns the paragraphs describing the pathogenesis of HE, and particularly the hyperammonemia. The main part of blood ammonium originates in the gut and is produced by intestinal bacteria. In dysbiosis, ammonia poduction may be increased, and endotoxin production has a lot of damaging effects.

In addition, due to the emergence of porto-systemic collaterals, part of ammonia bypasses the liver (not mentioned in the paper!).

  • Response 1: We appreciate the Reviewer’s thoughtful comment. We wrote additional content in the manuscript.

“Ammonia is continuously produced in human organs as a by-product of the catabo-lism of amino acids. In a healthy person, ammonia is produced by gut microbiota, and is converted to glutamine by the liver, thus preventing the penetration of ammonia into the systemic circulation [34]. In the liver, ammonia is converted to urea through the urea cycle in periportal hepatocytes, which is then excreted or eliminated through the urine and co-lon [35]. The urea cycle undergoes five major steps and requires enzymes including N-acetyl-glutamate synthase, carbamoyl phosphate synthetase, ornithine transcarbamyl-ase, argininosuccinate synthetase, argininosuccinic acid lyase, and arginase [36]. Among them, glutamine synthase is present in periportal hepatocytes and prevents the release of residual ammonia into the systemic circulation [37]. This enzyme synthesizes glutamic acid and ammonia and converts them to glutamine. However, in cirrhosis or equivalent liver failure, ammonia clearance decreases and progresses to hyperammonemia [38]. Ad-ditionally, when portal hypertension develops, various portal-systemic shunts occur as a complication, which is frequently observed in cirrhosis patients [39,40]. These por-tal-systemic shunts play a role in lowering portal pressure, but through this, normal liver flow is bypasses, resulting in increased blood concentrations without detoxification or metabolization of various toxic substances, including ammonia [41,42].”

  • Comment 2: Further point: Ammonia clearence by the liver is impaired (urea cycle in periportal hepatocytes, glutamin synthesis in perivenous cells. This an important nechanism, not mentioned in the paper. Considering these facts, Figure 1 must be improved.
  • Response 2: We added additional content to the manuscript and reflected the correction in Figure 1.
  • Comment 3: Next point: In Fig. 1, the most important causes of liver cirrhosis of chronic liver disease are mentioned. However, metabolic and autoimmune liver diseases are forgotten. Please correct.
  • Response 3: As the reviewer pointed out, we reflected the correction in Figure 1.
  • Comment 4: The presentation of studies about the link between gut dysbiosis and HE seems to be exhaustive. Maybe, it could be shortened. In page 4, several studies or authors are mentioned, but the citation lacks. 
  • Response 4: This is a very important point that we totally agree with. Citations may have been lacking because we summarized the contents of the studies we cited. We added some citations.
  • Comment 5: I know that the authors focus on gut dysbiosis. Therefore, multiple studies are indicated that are potentially effective in treatment of HE, mainly by by reducing hyperammonemia. However, the authors don't mentione other measures to lower ammonium levels, e.g. branched chain aminoacids, or L-ornithin-L-aspartate or similar compounds.

My question: Can nutrition alter dysbiosis in liver cirrhosis? The authors may comment on that.

  • Response 5: Thank you for providing us with nutritional options and what to include in additional treatment strategies. We have added the points you mentioned to the content. As our manuscript focused on the relationship between the gut microbiota and hepatic encephalopathy, we did not write weak links in the elements.

“4.2.3. Dietary

Therapeutic strategies for HE have been to modulate the systemic inflammation, neurotoxic substances such as ammonia, and changes in the composition and environ-ment of the gut microbiota [90]. Additionally, implementing these strategies in conjunc-tion with a diet at the same time can play a very important role in treatment and man-agement [91]. It has been known through many reports that changes in eating habits are closely related to the gut microbiota [92]. These dietary adjustments and changes have beneficial effects on the gut microbiome and affect disease improvement. Likewise, it can alter the cascades that lead to cognitive impairment by regulating gut nitrogen metabo-lism and improving inflammation [93]. For the management of HE, recommendations for dietary changes are presented as key clinical guidelines, which can make a significant contribution to treatment and prevention. The American Association for the Study of Liver Diseases (AASLD) and The European Association for the Study of the Liver (EASL) Prac-tice Guidelines [1] for HE recommend that a high-calorie diet be implemented in patients with acute decompensation cirrhosis, and that HE patients are recommended to intake 35-40 kcal/kg of energy per day. In particular, lipids in the diet are beneficial to patients with HE, as they have been shown to have beneficial effects on gut microbiota and bowel transit time [15].

Because abnormal nitrogen metabolism plays an important role in the pathogenesis of HE, early studies suggested that protein intake should be reduced [94]. However, further studies have reported that it has no beneficial effect on the progression of HE and worsens the nutritional status of patients by exacerbating protein breakdown in muscle [95]. Rather, it has been found that normal protein intake is well tolerated in HE and is useful in en-suring sufficient substrates for energy synthesis and hepatocyte function [96]. As addi-tional nutrients, branched-chain amino acids, such as valine, leucine, and isoleucine, have been shown to prevent excessive protein catabolism and reduce ammonia levels in patients with HE [97]. There are still obstacles and limitations in the application of these dietary and nutritional treatment modalities for HE, and high-quality randomized con-trolled trials are required [98]. In addition, there is still very little research on the effect of diet and nutrition on the relationship between HE and the gut microbiota we are dealing with. If the positive factors of diet on this relationship are identified or proven in the future, it could be another option for HE treatment strategies.”

Reviewer 3 Report

The manuscript is a narrative review  of the gut microbiota in hepatic encephalopathy. It has all the shortcommings of a narrative review (lacks the description of search methodology, the literature referenced is incomplete and the quantitative evaluation of the results of referenced studies is impossible. This lowers the impact of the parts of the manuscript that are concerned with the treatment options. 

Overall the manuscript lacks novelty altogether and presented information is not complete. The contents are rather superficial and are missing the depth - authors could maybe focus on one part - changes in the microbiota, or interventions regarding the microbiota with the aim of correcting HE.

Title is adequate, however it should be made clear that the manuscript is a review

Abstract  should include the type of the manuscript - i.e. narrative review and also the aim - what is the aim of the manuscript, what should the reader expect from it.

Introduction - categorical statement about probiotics efficacy is inappropriate as the evidence is inconclusive

Section 2 - cirrhosis dysbiosis index should be mentioned

4.2.1 - critical papers on lactulose are missing, e.g. metaanalysis by Als-Nielsen, BMJ, 2004

4.2.2 - critical papers on FMT are missing e.g. Bloom, Hepatology communications 2022

4.2.3 - critical papers on rifaximin are missing - e.g. Kang APT 2017

Author Response

ijms-1826515

“The link between gut microbiota and hepatic encephalopathy”

Point-to-point responses to comments by the Reviewer 3

First of all, we would like to thank the Reviewer 3 for his/her comments, which helped us to improve this manuscript.

Specific Comments:

The manuscript is a narrative review of the gut microbiota in hepatic encephalopathy. It has all the shortcommings of a narrative review (lacks the description of search methodology, the literature referenced is incomplete and the quantitative evaluation of the results of referenced studies is impossible. This lowers the impact of the parts of the manuscript that are concerned with the treatment options. Overall the manuscript lacks novelty altogether and presented information is not complete. The contents are rather superficial and are missing the depth - authors could maybe focus on one part - changes in the microbiota, or interventions regarding the microbiota with the aim of correcting HE. Title is adequate, however it should be made clear that the manuscript is a review

Response: We appreciate the Reviewer’s thoughtful comment. We added some paragraphs and revised manuscript to increase novelty.

  • Comment 1: Abstract should include the type of the manuscript - i.e. narrative review and also the aim - what is the aim of the manuscript, what should the reader expect from it.
  • Response 1: We checked the problem you mentioned and mentioned the aim of the manuscript, etc.

“This narrative review will identifiedy factors negatively influencing the gut-hepatic-brain axis leading to HE in cirrhosis and explore their relationship with the gut microbiome. We will also focused on the evaluation of reported clinical studies on the management and improvement of HE patients with a particular focus on microbiome-targeted therapy.”

  • Comment 2: Introduction - categorical statement about probiotics efficacy is inappropriate as the evidence is inconclusive
  • Response 2: We have checked what you said and added and corrected it.

“Current management of HE consists of intestinal-centered therapy with lactulose and lactitol, and nonabsorbable antibiotics such as rifaximin and neomycin [10]. Additionally, studies are underway to apply the modulating effect of intestinal microbes through probi-otics and prebiotics to the therapeutic management of HE [11]. Through this, it is being considered as another alternative as it has confirmed the possibility of improving cogni-tive function while suppressing harmful gut bacteria and lowering ammonia absorption [12,13]. In addition, although the management of HE through pharmacological ap-proaches and nutritional methods is being studied [14,15], the link with the intestinal mi-croflora has not been clearly identified.  We will explore the association of HE with the gut microbiota based on the gut-liver-brain axis and evaluate the therapeutic management of HE with a focus on clinical research data. Through this, we review the future direction by identifying the link with the gut microbiota.”

  • Comment 3: Section 2 - cirrhosis dysbiosis index should be mentioned
  • Response 3: We have checked what you said and added and corrected it.

“A similar pattern has been observed for dysbiosis in these cirrhosis conditions, and some researchers have devised and used several indices for the diagnosis and prognosis of cir-rhosis. One index that measures the degree of dysbiosis in cirrhosis is the Cirrhosis Dysbiosis Ratio (CDR) [23]. This index reported by Bajaj is calculated as the ratio of the abundance of Lachnospiraceae, Ruminococcaceae, Veillonellaceae and Clostridiales Incertae sedis XIV to that of Bacteroidaceae and Enterobacteriaceae. This index also correlates with endo-toxin and represents negative ecological and gut functional effects. Another index is the Gut Dysbiosis Index (GDI) reported by Wang [24]. The higher this index, the more severe dysbiosis appears. In addition, recently Bajaj et al [25]. reported that the addition of a mi-crobiota index to an RNA or DNA model in the stool analysis of patients could be signifi-cantly added to the MELD score to predict the risk of hospitalization for liver cirrhosis.”

  • Comment 4: 4.2.1 - critical papers on lactulose are missing, e.g. metaanalysis by Als-Nielsen, BMJ, 2004

4.2.2 - critical papers on FMT are missing e.g. Bloom, Hepatology communications 2022

4.2.3 - critical papers on rifaximin are missing - e.g. Kang APT 2017

  • Response 4: We have checked the important reference papers you mentioned and added and corrected the contents of the manuscript.

Reviewer 4 Report

There are a number of recent reviews of the links between the gut microbiota and hepatic encephalopathy; this review needs some refinement in both language and concepts; the authors need to provide a compelling reason in the introduction that is going to make their review stand out from other recently published reviews

There are times when the language is clumsy and makes it difficult for the reader to understand the concept/statement e.g. p2, line 15; p2, line 83 for a start 

There are a number of statements made in this review which have not been supported by a reference; please reference statements appropriately

Author Response

ijms-1826515

“The link between gut microbiota and hepatic encephalopathy”

Point-to-point responses to comments by the Reviewer 4

First of all, we would like to thank the Reviewer 4 for his/her comments, which helped us to improve this manuscript.

Specific Comments:

  • Comment 1: There are a number of recent reviews of the links between the gut microbiota and hepatic encephalopathy; this review needs some refinement in both language and concepts; the authors need to provide a compelling reason in the introduction that is going to make their review stand out from other recently published reviews
  • Response 1: We have checked what you said and added and corrected it.

  • Comment 2: There are times when the language is clumsy and makes it difficult for the reader to understand the concept/statement e.g. p2, line 15; p2, line 83 for a start 
  • Response 2: We have checked what you said and added and corrected it.

  • Comment 3: There are a number of statements made in this review which have not been supported by a reference; please reference statements appropriately
  • Response 3: We have checked what you said and added and corrected references.

Round 2

Reviewer 2 Report

The authors have addressed all the issues. The manuscript was significantly improved. Now, I recommend publication

Author Response

Thanks for reviewer's opinion. 

Reviewer 3 Report

the manuscript was improved, I detect no obvious scientific or ethic problems, however all the shortcomings of a narrative review remain. The manuscript provides no novelty, narrative reviews should visualize a path towards new areas of research or provide a new aspect to consider regarding the topic of interest, which this manuscript does not do.

the new sections could use proofreading, some sentences have completely different meaning e.g. "The use of nonabsorbable disaccharides has significantly confirmed minimal and overt therapeutic effects for HE"

I suspect it should have been ""The use of nonabsorbable disaccharides has significantly confirmed effect for  minimal and overt  HE"

Author Response

ijms-1826515

“The link between gut microbiota and hepatic encephalopathy”

Point-to-point responses to comments by the Reviewer 3

First of all, we would like to thank the Reviewer 3 for his/her comments, which helped us to improve this manuscript.

Specific Comments:

  • Comment 1: the manuscript was improved, I detect no obvious scientific or ethic problems, however all the shortcomings of a narrative review remain. The manuscript provides no novelty, narrative reviews should visualize a path towards new areas of research or provide a new aspect to consider regarding the topic of interest, which this manuscript does not do

  • Response 1: Thanks for these suggestions. In order to confirm the problems of the manuscript you mentioned and to suggest a new research area, we have added reinforcement to the 'Future directions' section.

“For the treatment and management of HE, the scope is expanding with convergence [120]with new areas: First, as dysbiosis is induced in cirrhosis and it has been confirmed that this gut microbiota imbalance affects bacterial products and metabolites [68,121], re-lated multiomics studies are being conducted. Researchers are focusing gut microbiota of metabolic potential comparable to liver, and are studying the treatment of liver disease and related complications based on the gut-liver axis [122,123]. In introducing the concept of ‘functional metagenomics’, Li et al. [124] demonstrated that functional key members of the gut microbiome have a significant impact on the metabolism and health of the host. Other researchers have discovered altered immune pathways through a multiplex ap-proach to profiling of transcripts, metabolites, and cytokines in blood samples from cir-rhosis patients [125]. Going forward, we will go beyond generally focusing on single or target components such as short-chain fatty acids [126] to deepen our understanding of gut microbiota dynamics in relation to host physiology and pathology [127], exploring the broad and complex host-microorganism metabolic interactions in the superorganism. This may provide new inspiration for the treatment of cirrhosis and HE. The second ap-proach to new territory is AI. Recently, methods utilizing artificial intelligence and ma-chine learning algorithms are being applied to hepatology [120]. Machine learning is the use of artificial intelligence to create and utilize predictive models on large data sets more efficiently and effectively than traditional methods [120]. MHE can be detected by special, time-consuming psychometric tests. Recently, as the diagnostic approach of MHE through artificial intelligence and machine learning is progressing, uncertainty resolution is being studied [128]. Artificial intelligence and machine learning are being applied in a variety of ways, from analysis using magnetic resonance imaging data [129,130]to methods of dis-tinguishing patients with different cirrhosis severities by the microbiome of saliva and fe-ces [65]. In particular, the use of artificial intelligence (AI) could become a necessity in the near future, especially for the utilization of large data sets, such as linking gut microbiome data and human biometric data. Artificial intelligence and machine learning will help classify disproportionate medical data and build risk prediction and diagnostic evalua-tion systems for liver cirrhosis with HE [131]. The third is a personalized therapeutic ap-proach. The gut microbiome has the potential to uniquely identify an individual like a fingerprint [132]. Therefore, even the same treatment may have different responses [133]. A method for determining baseline microbiome characteristics has been proposed to be con-sistent with an appropriate microbiome therapy. They also reported that a personalized approach to microbiome treatment based on baseline community structure and function may achieve the most clinical success [133].

The future direction of the treatment and management of cirrhosis and HE will be the link between the gut microbiota and the host based on the gut-liver axis.”

  • Comment 2: the new sections could use proofreading, some sentences have completely different meaning e.g. "The use of nonabsorbable disaccharides has significantly confirmed minimal and overt therapeutic effects for HE" I suspect it should have been ""The use of nonabsorbable disaccharides has significantly confirmed effect for  minimal and overt  HE"

  • Response 2: We have edited it to take into account the meaning of the sentence you said.

Reviewer 4 Report

The language remains clumsy in parts and the manuscript would benefit great from editing by an English 1st language editor

Many statements remain unreferenced

Please use current guidelines when describing dietary requirements

Author Response

ijms-1826515

“The link between gut microbiota and hepatic encephalopathy”

Point-to-point responses to comments by the Reviewer 4

First of all, we would like to thank the Reviewer 4 for his/her comments, which helped us to improve this manuscript.

Specific Comments:

The language remains clumsy in parts and the manuscript would benefit great from editing by an English 1st language editor. Many statements remain unreferenced. Please use current guidelines when describing dietary requirements

Response: On behalf of my team, I convey my best gratitude to Reviewer 2 for his/her comments, which helped us to improvise this manuscript. We did English Editing service at the first revision.  

  • Comment 1: Corrections on page 1 - The defining factor of HE is altered cognition
  • Response 1: We have added and edited what you said.

  • Comment 2: Corrections on page 3 - HE is not just a factor of advanced liver failure but is present in patients with decompensated cirrhosis and is associated with disease progression
  • Response 2: We have added and edited what you said.
  • Comment 3: Corrections on page 11 – Referece needed
  • Response 3: We added references to the section you mentioned.

  • Response 4: These EASL guidelines are old; please use current guidelines; current guidelines suggest 30-35kcals/kg/d and 1.2-1.5g protein/kg/dÚre the AASLD practice guidelines the ISHEN (International Society for Hepatic Encephalopathy and Nitrogen Metabolism) guidelines ?
  • Comment 4: We are grateful to the reviewer for this valuable and reasonable comment. We have updated the instructions in the part you mentioned. In addition, we have added content by attaching guidelines from other organizations.
